# Effect of *Sodium Hyaluronate* on Antioxidant and Anti-Ageing Activities in *Caenorhabditis elegans*

**DOI:** 10.3390/foods12071400

**Published:** 2023-03-26

**Authors:** Qianmin Lin, Bingbing Song, Yingxiong Zhong, Huan Yin, Ziyu Li, Zhuo Wang, Kit-Leong Cheong, Riming Huang, Saiyi Zhong

**Affiliations:** 1College of Food Science and Technology, Guangdong Ocean University, Guangdong Provincial Key Laboratory of Aquatic Product Processing and Safety, Guangdong Province Engineering Laboratory for Marine Biological Products, Zhanjiang 524088, China; 20222145022@stu.scau.edu.cn (Q.L.); 15891793858@163.com (B.S.); yinhuan741852@163.com (H.Y.); lzywjp833@163.com (Z.L.); wangzhuo4132@outlook.com (Z.W.); klcheong@gdou.edu.cn (K.-L.C.); 2College of Food Science, South China Agricultural University, Guangzhou 510642, China; 3Collaborative Innovation Center of Seafood Deep Processing, Dalian Polytechnic University, Dalian 116034, China; 4Shenzhen Research Institute, Guangdong Ocean University, Shenzhen 518108, China

**Keywords:** *Sodium Hyaluronate*, *Caenorhabditis elegans*, anti-ageing, oxidative stress, antioxidant enzymes

## Abstract

As an acidic polysaccharide, the formation of *Hyaluronic acid* (HA) is typically *Sodium Hyaluronate* (SH) for knee repair, oral treatment, skincare and as a food additive. Nevertheless, little information is available on the anti-ageing activity of SH as a food additive. Therefore, we treated *C. elegans* with SH, then inferred the anti-aging activity of SH by examining the lifespan physiological indicators and senescence-associated gene expression. Compared with the control group, SH (800 μg/mL) prolonged the *C. elegans*’ lifespans in regular, 35 °C and H_2_O_2_ environment by 0.27-fold, 0.25-fold and 1.17-fold. Simultaneously, glutathione peroxidase (GSH-Px), antioxidant enzyme superoxide dismutase (SOD) and catalase (CAT) were increased by 8.6%, 0.36% and 167%. However, lipofuscin accumulation, reactive oxygen species (ROS) and malondialdehyde (MDA) were decreased by 36%, 47.8–65.7% and 9.5–13.1%. After SH treatment, athletic ability was improved and no impairment of reproductive capacity was seen. In addition, SH inhibited the blocking effect of *age-1* and up-regulated gene levels involving *daf-16*, *sod-3*, *gst-4* and *skn-1*. In conclusion, SH provides potential applications in anti-ageing and anti-oxidation and regulates physiological function.

## 1. Introduction

*Hyaluronic acid* (HA) is an acidic mucopolysaccharide that widely presents in the human body. As a vital component of the extracellular matrix, HA performs essential physiological functions, including moisturizing the skin, lubricating knee joints [1], facilitating cell proliferation and metastasis, etc. In aqueous HA solution, the anionically charged carboxyl groups usually combine with metal ions such as Na^+^, then form the high-hydrophilic *Sodium Hyaluronate* (SH) [2]. Based on its good water-holding and rheological properties, HA improves the texture of yogurt [3], enhances the homogeneity of smoked sausages [4] and contributes to the stability of skimmed milk in combination with carrageenan [5]. Interestingly, Kweon et al. used HA to produce an edible film for alleviating dry mouth [6]. Notably, countries have published standards for the usage of HA. In 2021, China officially approved SH as a new resource food, which can be used as a raw material for health food. Before that, the USA, Korea and Japan had already issued standards for HA to be used as a food additive or functional food [7]. Importantly, many articles have reported on the safety of oral HA. Balogh et al. found that Wistar rats and Beagle dogs fed SH were largely excreted without accumulation in vivo [8]. It was reported that oral administration of SH does not cause genotoxicity, acute and subchronic toxicity in rats [9], and no side effects were found with a 90-day feeding [10]. Besides the food sector, SH has been used in biomedical applications, including knee pain, skincare, dry eye, anti-wrinkle [11], dental [12] and drug delivery [2]. Nevertheless, SH has been used as a food additive for a long time and its safety, antioxidant and anti-ageing activities still need to be studied in depth.

*Caenorhabditis elegans* (*C. elegans*) are easily cultured, simply manipulated and usually explore the antioxidant and anti-ageing effects of drugs. Based on a nematode model, purple pitanga fruit enhances resistance to H_2_O_2_ oxidative stress and inhibits ROS [13]. *Blumen laciniata* polyphenols [14] and orange extracts [15] promote an anti-stress response and prolong lifespan. In addition, *C. elegans* have been entirely sequenced, and 60–80% of genes are high-homologous to humans. Signaling pathways affecting ageing are intensively studied, such as insulin/insulin-like growth factor (IIS), a target of rapamycin (TOR), dietary restriction, AMPK and sirtuins-2. Therefore, it is convenient to explore the antioxidant and anti-ageing effects of SH.

Overall, this paper examined the effects of SH on nematodes in terms of longevity, physiological indicators, environmental stresses and gene expression. The physiological indicators included fecundity, motility, lipofuscin, reactive oxygen species (ROS), superoxide dismutase (SOD), catalase (CAT), malondialdehyde (MDA) and glutathione peroxidase (GSH-Px). The environmental stress experiments contained 35 °C heat shock and H_2_O_2_ oxidative damage.

## 2. Materials and Methods

### 2.1. Materials and Reagents

Wild-type *C. elegans* (N2) and uracil leakage defective *E. coil OP50* (OP 50) were obtained from the *C. elegans* Genetics Center (CGC). SH (rooster comb) was purchased from Shanghai Macklin. GSH-Px and CAT kits were purchased from Suzhou Grace. SOD and MDA kits were purchased from Nanjing Jiancheng. 2,7-dichloro-dihydro fluorescein diacetate (DCFH-DA) and total protein (BCA) kits were purchased from Beyotime. 5-fluoro-2-deoxyuridine (FUDR) was purchased from Shanghai Yuanye. Thirty percent H_2_O_2_ was purchased from Guangzhou Chemical Reagent Factory. Other reagents and solvents were of analytical grade.

### 2.2. Drug Sensitivity Test

According to previous reports [16,17], The drug sensitivity test was carried out with a slight adjustment. SH was diluted with ultrapure water to 0, 200, 400 and 800 μg/mL. Small circular papers (d = 5 mm) were fully immersed in SH solution, then air dried. OP50 (400 μL) was spread evenly on LB agar culture (90 mm). Then, the papers were stuck after the OP50 dry. The mediums were incubated in the thermostatic incubator (Shanghai Yiheng) at 37 °C and we observed the growth change of colonies around the film after 24 h.

### 2.3. Lifespan and Fundamental Physiological Indicators

#### 2.3.1. Lifespan Assay

Synchronized L4 period *C. elegans* were selected and transferred to the nematode growth medium (NGM) containing FUDR (150 μM), OP50 (200 μL) and SH (0, 200, 400 and 800 μg/mL). Three plates were set up for each concentration group with 35 worms per plate and incubated at 20 °C. Every 48 h, nematodes were transferred to the new NGM with fresh food. The number of nematodes surviving, dying and lost was recorded regularly every day until all nematodes were dead. Lifespan assays were repeated three times.

#### 2.3.2. Fecundity

The nematodes were selected and incubated for 14 h in the L1 period, then transferred to NGM containing OP50 and SH. Five worms were selected for each experimental group, one worm per dish. The plates containing the offspring were incubated at 20 °C for 24 h, then the number of offspring was recorded. The plates were changed daily at regular intervals until the end of the spawning period.

#### 2.3.3. Motility

For the lifespan assay, the nematodes’ motilities were recorded on the 7th, 10th and 14th days of administration. According to Koch’s method [18], with a small adjustment, the nematode locomotor behaviors were graded as follows. A normal nematode was one that moved quickly, avoided external disturbances and its trajectory resembled a sine curve, but the sluggish one did not follow a sinusoidal trajectory. Within 3 s of external disturbance, a worm that wobbled slightly was defined as immobile, otherwise dead.

#### 2.3.4. Lipofuscin

L4 period *C. elegans* were selected, and 60 nematodes were divided equally into four experimental groups and incubated at 20 °C for 96 h. Plates were changed every other day. At 96 h, the nematodes were cleaned by M9 buffer [19], then transferred to NGM without food. A small drop of NaN_3_ (20 μM) was added to anaesthetize the nematodes. After the worms were free of oscillations, they were quickly transferred to slides containing 1% agarose gel and observed by an inverted fluorescence microscope (ThermoFisher, Shanghai, China). All pictures were magnified 100 times, and the nematodes’ in vivo relative fluorescence intensity was measured by Image J software.

### 2.4. Stress Assay

The L4 period *C. elegans* were incubated at 20 °C for 96 h and changed every other day.

#### 2.4.1. Heat Shock

At 96 h, the worms were transferred to blank NGM with 30 nematodes per medium and incubated at 35 °C. The number of nematode deaths was recorded every 2 h until all nematodes were completely dead.

#### 2.4.2. Oxidative Stress

Referring to the method of Wang et al. [20], after 96 h, nematodes were transferred to NGM containing 0.3% H_2_O_2_, then put back at 20 °C. The number of nematode deaths was recorded every hour until the worms were utterly dead [21].

### 2.5. ROS Level, Antioxidant Enzyme Activities and MDA Content

After 96 h SH treatment, approximately 800 nematodes per group were collected in Ep tubes. The worms were repeatedly rinsed with M9 buffer, and the supernatant was discarded by centrifuging (ThermoFisher, Shanghai, China) at 3000 rpm for 2 min. Next, 400 μL saline was added to each tube and placed on ice. Finally, the worms were crushed by using a cellular ultrasonic crusher (Scientz, Ningbo, China) at 200 W for 10 min.

#### 2.5.1. ROS Level

The worm suspension was centrifuged at 3000 rpm for 2 min, then 50 μL of supernatant and 50 μL of DCFH-DA (100 μM) per well were added to a 96-well black plate. Immediately afterwards, the fluorescence intensity was measured every 5 min for 2 h at 37 °C using an enzyme marker (ThermoFisher, Shanghai, China), where the excitation wavelength was 485 nm and the emission wavelength was 538 nm. Additionally, the experimental blank group was free of nematodes and fluorescence, and the control group was added with only nematodes and no fluorescent probes.

#### 2.5.2. Antioxidant Enzyme Activities and MDA Content

The BCA, MDA, SOD, CAT and GSH-Px were determined separately according to the kit instructions.

### 2.6. Quantitative Real-Time PCR (RT-qPCR)

After fine-tuning based on Yang’s method [22], the total RNA was extracted by crushing the nematodes and then adding 1 mL of TRIzol solution (Beyotime, Shanghai, China). As described in the Prime Script^TM^ RT kit (Takara, Beijing, China, the cDNA was obtained by reverse transcription reflection in a PCR instrument (Eppendorf, Shanghai, China). Following this, real-time PCR was completed using the iQ ^TM^ SYBR ^R^ Green Supermix kit and anti-proliferative factor expression was detected by the BIO-RADCFX48^TM^ real-time system. Lastly, the relative expression of the anti-proliferative genes was calculated based on the 2^ΔΔCt^. Real-time PCR primer sequences can be found in Appendix A.

### 2.7. Statistics

Data and graphs were processed by GraphPad Prism 9.0 and the log-rank tests were analyzed by IBM SPSS Statistics 26. Image J software was used to measure the lipofuscin relative fluorescence intensity. All data were expressed as mean ± SD. The values expressed as percentages (%) were standardized to 100% for the control group. The asterisks (*) were considered to be significantly different (ns = no significant difference, * *p* < 0.05, ** *p* < 0.01, *** *p* < 0.001, **** *p* < 0.0001).

## 3. Results

### 3.1. SH Has No Antimicrobial Activity against OP50

See Appendix A for drug sensitivity testing. No significant inhibition circles were formed around the edges of the SH-containing paper sheets, so it can be assumed that the growth and reproduction of OP50 were uninhibited by SH.

### 3.2. Lifespan and Fundamental Physiological Indicators

#### 3.2.1. SH Extends the Lifespan of *C. elegans*

The nematodes’ lifespan is shown in Figure 1a, where the mean lifespan of control nematodes was 15 days. Compared with the control group, the average lifespan of nematodes after treatment with different concentrations of SH (200, 400 and 800 μg/mL) was 17, 18 and 19 days, with extensions of 13.3%, 20.0% and 26.7%, respectively. It can be seen that SH prolonged the worm lifespan and the survival curve of the experimental group shifted to the right, showing a concentration-dependent longevity prolongation effect in the experiment.

#### 3.2.2. SH Shows No Toxicity on *C. elegans* Fecundity

As shown in Figure 1b, the worm spawning period was about 4–5 days, mainly concentrated in the first 3 days. The nematodes in the experimental group were fed SH continuously from the L4 stage to the end of parturition, but no significant difference was found in the total number of offspring between the groups. It is reasonable to assume that the reproductive capacity of nematodes is not impaired by SH and provides evidence for the safety of SH.

#### 3.2.3. SH Slows down the Decline of *C. elegans* Motility

The effect of SH on nematode motility was recorded on days 7, 10 and 14 after dosing, and the results are referred to in Figure 1c. No worms died during the first 7 days. On day 10, nematode motility gradually decreased and the number of sluggish nematodes in the control group increased from 6.5% to 17.2%. After 14 days of treatment, the proportion of normal nematodes in the control group was 22.6%, while those receiving SH (200, 400 and 800 μg/mL) were 1.89, 2.33 and 2.69 times higher; additionally, their motility was better than that of the control group. In contrast, the SH-treated groups had a lower proportion of sluggish and immobile nematodes, with 32.3%, 29.2%, 20.9% and 15.2% of sluggish worms and 17.2%, 13.5%, 13.2% and 9.8% of immobile worms in each group, respectively. Thus, nematode movement decreased progressively with increasing longevity, but SH slowed the decline in nematode motility.

#### 3.2.4. SH Reduces Lipofuscin Level in *C. elegans*

Lipofuscin is a sign of ageing in organisms. Previous studies have shown that lipofuscin accumulates in nematodes and that fluorescence increases gradually [23]. Under an inverted fluorescence microscope, the green fluorescence of lipofuscin in nematodes can be observed. Compared with the control, SH treatments at 200, 400 and 800 μg/mL reduced lipofuscin by 14.7%, 24.6% and 36.4%, Figure 2. It can be concluded that SH reduces the formation of lipofuscin in sibling senescent nematodes.

### 3.3. SH Enhances the Abilities of C. elegans to Resist Stress

#### 3.3.1. SH Relieves Heat Shock

The nematodes were incubated in a temperature 35 °C above the optimum physiological temperature. Similar to other studies [24,25], the average lifespan of the control group was 8 h, and the heat shock model was successfully produced as shown in Figure 3a. The nematode resistance to heat was enhanced by the SH treatment, with a maximum lifespan of 16 h and a 25% longer average lifespan. It can be hypothesized that SH has a protective effect against heat stress injury and enhances nematode health by increasing heat tolerance.

#### 3.3.2. SH Protects *C. elegans* during Oxidative Stress

The results of H_2_O_2_ oxidative damage to nematodes are shown in Figure 3b. The controls had a mean lifespan of 6 h and a maximum lifespan of 14 h. In contrast, the mean lifespan after SH treatment (200, 400 and 800 μg/mL) was 8, 8 and 13 h, respectively, with a maximum lifespan of 18 h. Noticeably, nematodes in the SH-treated group (800 μg/mL) did not show mortality until the 3rd h, with survival dropping sharply at the 12th h. We can predict that the oxidative damage suffered by nematodes in the H_2_O_2_ environment can be mitigated by SH.

### 3.4. SH Decreases ROS Level in C. elegans

As observed in Figure 4a,b, the control nematodes had the highest ROS levels. SH (200, 400 and 800 µg/mL) treatment lowered the relative levels of ROS by 0.657-fold, 0.429-fold and 0.478-fold, respectively. Excess ROS has been shown to damage the body, leading to disease and life-threatening conditions [26]. From the downward shift in the ROS content curves, we speculate that ROS may be reduced in nematodes by SH treatment, thus extending their lifespans.

SH increases antioxidant enzyme activities and reduces MDA content. Figure 4c–f show the effects of SH on the antioxidant enzymes and MDA content in nematodes. The activity of CAT enzymes in nematodes was increased by 0.24-fold, 1.18-fold and 1.67-fold after SH (200, 400 and 800 μg/mL) treatment, compared with the control group. After nematodes were given SH (800 μg/mL), the levels of SOD and GSH-Px increased by 36% and 8.6%, while the other SH concentration (200, 400 μg/mL) groups were not significantly different from the controls. In addition, MDA levels in the SH-fed nematodes were reduced by 9.5–13.1% compared with the control group. It has been suggested that CAT has oxidative damage defense, SOD has the ability to scavenge free radicals to prevent cell damage and ageing-pigment and GSH-Px has a peroxide-degrading effect [27,28]. After SH treatment, nematode antioxidant enzyme activities were increased, and the levels of oxidative damage were reduced. Interestingly, Ke et al. indicated that in vivo antioxidant enzyme activities were increased by low molecular weight HA in immunosuppressed mice [29]. In summary, we can expect that SH has a good performance on antioxidant and anti-ageing effects in nematodes.

### 3.5. SH Regulates the mRNA Expression of C. elegans

The insulin/insulin-like growth factor signaling (IIS) pathway plays a crucial role in human ageing and disease. We found that SH prolongs the lifespan of nematodes, regulates physiological functions and produces good antioxidant effects. Therefore, the expressions of genes associated with the IIS pathway were further examined by rt-qPCR. Figure 5a shows that there was no significant difference in the expression of *daf-2* between the groups, but age-1 expression decreased by 5–21.9% with increasing concentrations of SH treatment. The levels of *daf-16* increased by 10.6%, 14.6% and 29.1% after SH treatment (200, 400 and 800 μg/mL) compared with the control group. Compared with the control group, *daf-16* levels increased by 10.6%, 14.6% and 29.1% after SH treatment (200, 400 and 800 μg/mL). The relative expressions of *sod-3* and *gst-4*, the downstream oxidative genes of *daf-16*, also increased by 5.4% to 15.0% and 5.2% to 17.7%. Further, the relative expression of *skn-1* in the MAPK oxidative stress signaling pathway increased by 0.129-fold to 0.179-fold after SH treatment. In summary, *daf-16*, *sod-3*, *gst-4* and *skn-1* were upregulated by SH treatment, but age-1 was downregulated, Figure 5b.

## 4. Discussion

SH is the sodium salt form of HA. The metabolic safety and functionality of HA and SH were extensively studied by the Japanese in 1990. HA and its derivatives have long been used in various fields. When synovial fluid in joints is reduced, SH injections can relieve joint friction damage [30]. SH has also been made into cosmetics for skin moisturization and anti-wrinkle effects. In recent years, researchers have focused on SH as a vehicle for targeted drug delivery [31], modulation of inflammation [32] and other directions. With the development of HA and SH in the food sector gaining international attention, we have also carried out studies on the antioxidant and anti-ageing activities of SH using nematode organisms.

The average nematode lifespan was prolonged by SH treatment (800 μg/mL) by 26.7% compared with the control group. According to the principle of energy conservation, substances that extend the life span of nematodes reduce their fecundity [33]. However, we found that SH did not affect the nematode fecundity, and on the one hand SH had a protective effect on post-reproductive maternal nematodes [34]. In times of food shortage, maternal nematodes adapt to their environment by shortening their lifespan or reducing the number of offspring [35]. On the other hand, we ensured that food was adequate and no inhibitory effect of SH on OP50 growth was detected in drug sensitivity tests, thereby reducing the nematode survival threat [36]. It has long been reported that SH injections do not affect the parents and offspring of rats [37] and rabbits [38]; the maximum non-teratogenic dose of SH is 20 mg/kg in rabbits [39] and 50 mg/kg/day in rats [40]. SH is commonly used to treat degenerative knee osteoarthritis, and the motility of SH-treated senescent nematodes was significantly better than that of the controls. Activation of *daf-16* enhanced mitochondrial muscle mass and activated *daf-2* to strengthen muscle function and motility [41], so it can be hypothesized that activation of *daf-16* led to increased motility in older nematodes [42].

The nematodes treated with SH increased their average lifespan by 25% under heat shock at 35 °C. Under oxidative stress with H_2_O_2_, SH (800 μg/mL) obviously slowed nematode mortality and increased their average lifespan from 6 h to 13 h. It has been shown that when heat and oxidative stress are applied to nematodes simultaneously, heat stress inhibits oxidative stress and affects their respective outcomes [43]. The therapeutic effect of SH when nematodes are exposed to both heat and oxidation is worthy of further study later. After nematodes were given SH (800 μg/mL), ROS and CAT contents were increased by 0.478-fold and 1.67-fold and the levels of SOD and GSH-Px increased by 36% and 8.6%, while MDA levels were reduced by 13.1% compared with the control group. SH has a similar antioxidant capacity to natural fruits and vegetables; these can enhance nematode antioxidant enzyme activities, reducing free radical accumulation cellular injury [15,19]. Among them, capsaicin [44] and cashew leaves [45] can also reduce ROS levels and increase SOD and CAT enzyme activities by stimulating daf-16 and skn-1, thereby regulating antioxidant capacity and extending lifespan. Thus, we hypothesized that SH could increase CAT, SOD and GSH-Px antioxidant enzyme activities, reduce ageing markers such as lipofuscin, ROS and MDA, and then enhance tolerance to abnormal environments and prolong lifespans.

The effect of SH in the IIS signaling pathway was predicted by RT-qPCR, with no significant change in the upstream gene *daf-2*. However, *age-1* expression was reduced after SH treatment, but we hypothesize that it was SH that inhibited this blockade. Then, *daf-16* expression was increased and downstream *sod-3* and *gst-4* genes were also activated, with increased nematode resistance and longevity. Sour cherry [46] and blueberry-apple [47] also prolonged nematode lifespan in a *daf-16*-dependent manner, as did gallocatechol (EGCG), which upregulated *sod-3* expression [48] and enhanced stress resistance. *skn-1* is a crucial antioxidant gene in AMPK, rose oil [49] and black mulberry [50], which activate both *skn-1* and its downstream gene *gst-4*, not only act as antioxidants but also delay Alzheimer’s disease. It is conjectured that SH can enhance the antioxidant capacity and prolong the lifespan of nematodes by increasing the levels of *skn-1* and *gst-4*.

## 5. Conclusions

This study showed that SH prolonged the lifespan of *C. elegans* and significantly improved their motility without causing damage to their reproductive capacity. SH also enhanced the resistance of nematodes under heat and oxidative stress. It significantly reduced the ROS content, decreased senescence pigments and increased antioxidant enzyme activities in nematodes, thus enhancing their antioxidant capacity and prolonging lifespan. In the present study, it was initially found that *daf-16*, *sod-3*, *gst-4* and *skn-1* were upregulated by SH treatment, while *age-1* was downregulated. Whether SH acts on other signaling pathways to synergistically enhance lifespan extension remains to be analyzed. However, the available studies provide further evidence of the safety of SH use and its anti-ageing and antioxidant efficacy.

## Figures and Tables

**Figure 1 foods-12-01400-f001:**
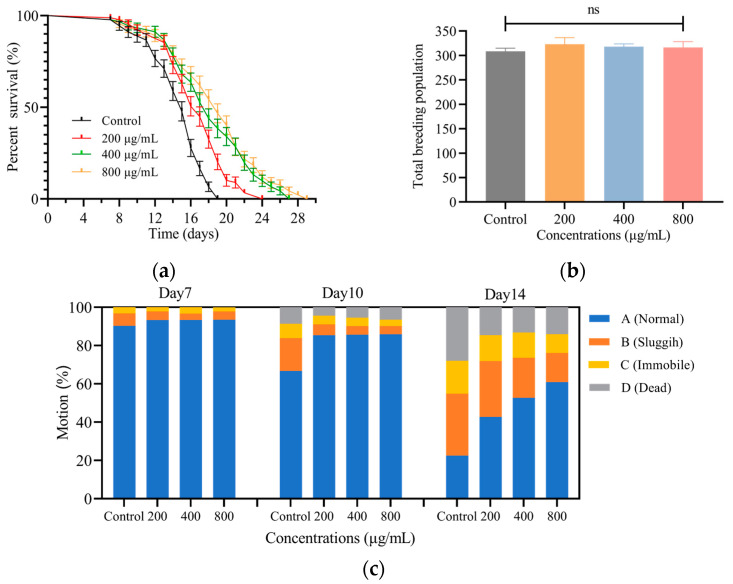
(**a**) SH prolonged the lifespans of *C. elegans*. The black curve represent as the control group and the colored curves represent the SH administration groups. The GraphPad Prism 9.0 survival curve function was used to analyze the growth of nematodes. (**b**) The reproductive capacity of *C. elegans* was not found to be impaired by SH. There was no significant difference in the number of nematode offspring treated with SH compared with the control group. (**c**) SH slowed down the decline in motility of *C. elegans*. The number of sluggish and immobile nematodes gradually increased in the control group, but the decline of normal nematodes was slowed down by SH treatment. Three independent experiments were carried out. Notes: ns meant not significant.

**Figure 2 foods-12-01400-f002:**
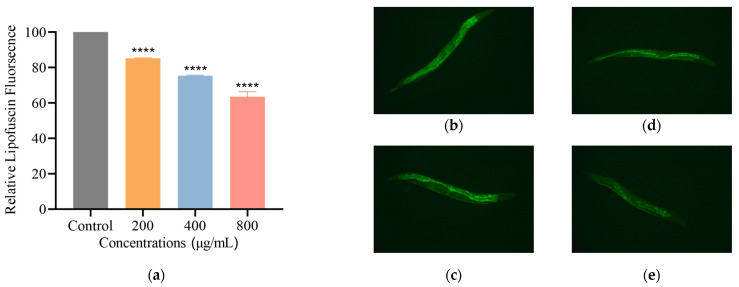
SH reduced senescence pigments in nematodes of the same age. Lipofuscin in nematodes is an auto-fluorescent pigment that can be observed by inverted fluorescence microscopy, and the accumulation of senescence pigments was analyzed by Image J software. (**a**) shows the relative fluorescence intensity of lipofuscin in each group of nematodes. (**b**–**e**) show the control group, and the 200 μg/mL, 400 μg/mL, 800 μg/mL SH treatment groups, respectively, as observed under inverted fluorescence microscopy. Lipofuscin assays were carried out in triplicate. Notes: The asterisk (****) represents the significant difference; the larger the number, the more significant.

**Figure 3 foods-12-01400-f003:**
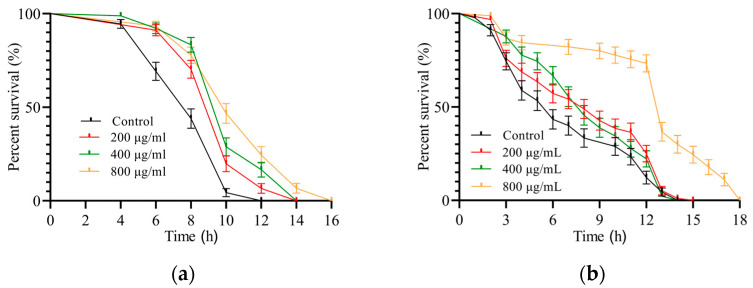
The black curve is the control group and the colored curves are the SH-treated groups. The colored curves in both graphs are shifted to the right, indicating that SH prolongs the lifespan of nematodes under stressful conditions. (**a**) is the survival curve of nematodes under heat shock and (**b**) is the survival curve under H_2_O_2_ oxidative stress. Each stress experiment was conducted three times.

**Figure 4 foods-12-01400-f004:**
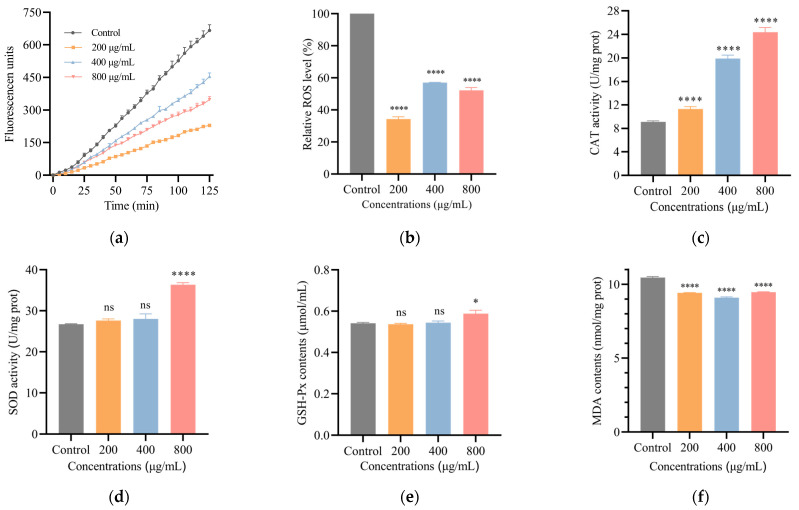
Groups (**a**,**b**) show the real-time changes in ROS and the maximum relative levels of ROS after 2 h. (**c**–**e**) show the CAT, SOD and GSH-Px antioxidant enzyme activities in nematodes. (**f**) shows the MDA content in nematodes. Three independent experiments were carried out for all indicators. Data are shown in mean ± SD, n = 3. Notes: The asterisk (* and ****) represents the degree of significant difference. The higher the number, the more significant; ns means not significant.

**Figure 5 foods-12-01400-f005:**
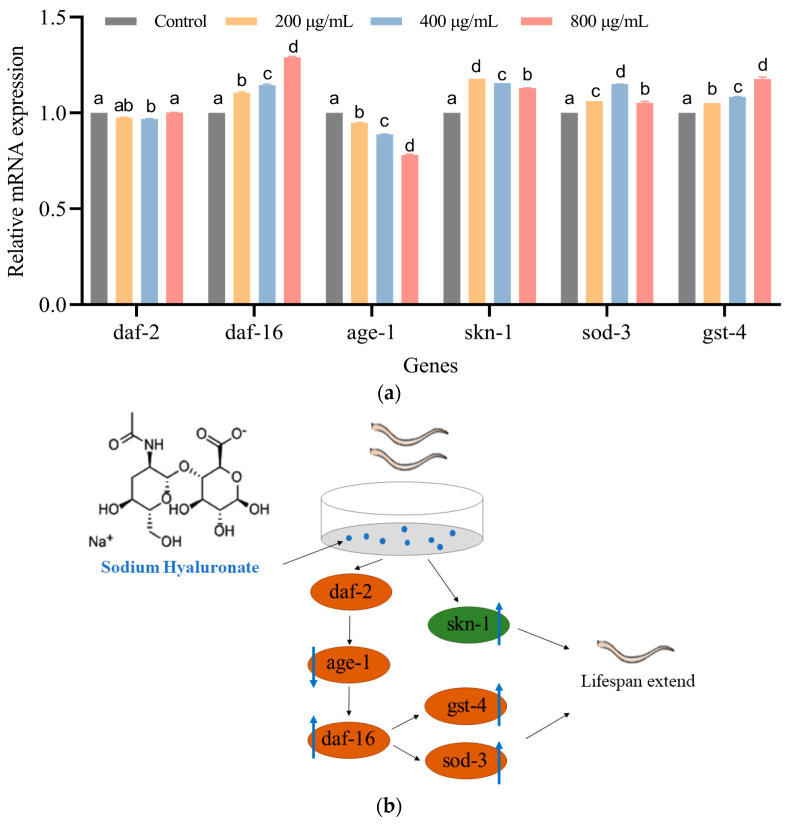
(**a**) Effect of SH on the expression of *C. elegans*-related genes. (**b**) Possible molecular mechanism of the anti-ageing effect of SH. Data are shown in mean ± SD, n = 3. Notes: no common letter in the same gene column indicates a significant difference (*p* < 0.05). Additionally, ns represents not significant.

## Data Availability

Data are contained within the article and Appendix A.

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
