# Peer review of "Effect of *Sodium Hyaluronate* on Antioxidant and Anti-Ageing Activities in *Caenorhabditis elegans"

_foods, 2023, doi:10.3390/foods12071400_

Round 1

Reviewer 1 Report

The presented paper investigates the effect of sodium hyaluronate on lifespan, fecundity and motility and resistance under heat and oxidative stress of C.elegans by measuring their antioxidant capacity, antioxidant enzymes activity and senescence pigments accumulation. The scientific soundness and practical importance of this paper is satisfactory, but I have a couple of serious objections:

It is not clear to me how the topic of this paper fits into the scientific field of the Foods, journal of the food science? In the introduction and abstract it is mentioned that hyaluronic acid  is used in food processing. Line38-40. Explain in more detail the application of SH as a food ingredient and give examples.

Relate how this research contributes to food science or choose another journal for submission.

Line 44-45 Latin terms in vitro and in vivo should be italicized

Provide information on whether in vivo research on the effects of hyaluronic acid on nematodes or on some other model organisms has already been done in previous research by other authors.

Line 79-81 Do not use the imperative but the passive.

In materials and methods section, the devices used and their type, manufacturer, the place of production, such as incubators, centrifuges, spectrophotometers, are not listed.

Line 210-213; 272-274; 276-278; 291-292 Cite literature references from where it was taken.

Line 338-341 Compare the results obtained in the aforementioned works with the results of this research.

Author Response

Dear reviewer, thank you very much for your constructive suggestions! Here are our responses and corrections. Details are attached.

Question 1: It is not clear to me how the topic of this paper fits into the scientific field of the Foods, journal of the food science? In the introduction and abstract it is mentioned that hyaluronic acid is used in food processing. Line38-40. Explain in more detail the application of SH as a food ingredient and give examples.

Respond: Thank you very much for your suggestions. Firstly, we have added the progress of research on Sodium Hyaluronate (SH) as a food additive (highlighted in line 37-41). Secondly, metabolic mechanisms, safety and reproductive toxicity experiments of SH studied on other model organisms are added (highlighted in line 44-48, line 311-313). Finally, based on C. elegans, we found that SH treatment has antioxidant and anti-ageing effects. With the above modifications, we hope that this study will be more closely aligned with the foods journal theme.

Question 2: Relate how this research contributes to food science or choose another journal for submission.

Respond: Thank you for your kind advices. We still hope to continue to submit to foods journal. On the one hand, we have added more information about SH researches in the food sector (highlighted in line 37-41). On the other hand, we found that the nematode lifespan was extended without impairing reproduction by SH treatment (highlighted in line 305-307). Last but not least, we sincerely hope that we can continue to contribute to foods.

Question 3: Line 44-45 Latin terms in vitro and in vivo should be italicized.

Respond: Thank you for the reminder, we have revised it and amended the text for the same situation (highlighted in line 46, line 114, and line 269).

Question 4: Provide information on whether in vivo research on the effects of hyaluronic acid on nematodes or on some other model organisms has already been done in previous research by other authors.

Respond: Thanks for this very constructive advice. We have added this information in the introduction (highlighted in line 44-48) and discussion (highlighted in line 311-313) section of the article.

Question 5: Line 79-81 Do not use the imperative but the passive.

Respond: Sorry for our mistakes, we have modified this language expression in revised manuscript (highlighted in line 80-82).

Question 6: In materials and methods section, the devices used and their type, manufacturer, the place of production, such as incubators, centrifuges, spectrophotometers, are not listed.

Respond: Sorry for our mistakes, we have added these information (highlighted in line 83-84, line 113, line 130, line 132, line 137).

Question 7: Line 210-213; 272-274; 276-278; 291-292 Cite literature references from where it was taken.

Respond: Thank you for the careful reminder, we have added the relevant content (highlighted in line 204-205, line 266-269).

Question 8: Line 338-341 Compare the results obtained in the aforementioned works with the results of this research.

Respond: Thank you for the reminder. We have made changes to this section (highlighted in line 319-321 line 323-327).

Finally, thank you very much for your constructive help and guidance!

Author Response

Dear reviewer, we thank the reviewer for your generous and enthusiastic comments. The suggestions on how to improve the manuscript are helpful, and here are our responses and corrections.

Question 1: The biggest drawback of the manuscript is the poor quality of English. There are innumerable grammatical errors, sentences with no meaning and incomplete sentences. This reduces the enthusiasm of the reviewer (and the readers)! This needs to be corrected and the manuscript needs to be checked/formatted preferably by native English speaker.

Respond: We apologized these mistakes. In the revised manuscript, we edited the manuscript carefully and made every effort to improve the English throughout the paper.

Question 2: A reasonably big list of compounds that extend C. elegans lifespan already exists. Sodium hyaluronate could just be an addition to that list but the manuscript doesn’t increase our understanding of how sodium hyaluronate is (if at all) different from other compounds and whether there is any specificity. The ‘specificity’ in this case is extremely important and I don’t find any effort to show that in the manuscript.

Respond: Thanks for this very constructive advice. We tried our best to clarify the ‘specialty’ of Sodium Hyaluronate (SH). On the one hand, we have added information on SH as a food additive in the introduction of the article (highlighted in line 37-41), to further verify the safety of SH by nematodes (highlighted in line 44-48). On the other hand, we have added in the discussion section of the article (highlighted in line 303-305, line 309-311) that SH injections do not have side effects on parents and offspring, SH treatment did not reduce nematode fecundity either. Finally, we clarified the theme of the study in the conclusion of the article (highlighted in line 349-352), which was to evaluate the safety, antioxidant and anti-ageing activity of SH through C. elegans.

Question 3: In multiple places, figure reference is missing (e.g., line 246, line 229, line 264 etc.)!

Respond: Thanks for reminding us. Figure references (highlighted in line 178, line 186, line 194, line 209, line 226-227, line 238, line 252, line 259, line 282 and line 291) are fully checked and corrected in revised manuscript.

Question 4: In majority of figures, the number of times each experiment was performed (usually represented as ‘n’) is missing.

Respond: Sorry for our mistakes, the number of times each experiment was added (highlighted in line 173-174, line 219, line235-36, line 249-250, line 276).

Question 5: In the text, figure 1c is described before figure 1b.

Respond: Thank you for the helpful reminder. In revised manuscript, we have switched the original positions of the two parts, the fecundity and the motility (highlighted in line 169-170, line 171-172, line 185 and line 193).

Question 6: Line 159: Supplementary figure reference is missing.

Respond: Sorry, we have added the figure legend (highlighted in line 163) to the revised manuscript.

Question 7: For fig 1b and 1c, figure legends are not matching the figures and text.

Respond: Very sorry. In response to this issue, we have corrected fig. 1(b) and 1(c) figure legends (highlighted in line 169-170, line 172-173, line 184 and line 191) and checked the other figure legends in the paper.

Question 8: For the lipofuscin experiment, the proper experiment would be to first show that lipofuscin increases with age by comparing its level in younger and older control animals and then show that this increase is not observed/less observed in presence of sodium hyaluronate.

Respond: We strongly agree with your suggestion that the data and figure provide more visual evidences of the accumulation of lipofuscin with increasing age. There have been many studies proving that ageing causes lipofuscin to accumulate, so we have added relevant references first (highlighted in line 204-205). Then, after many iterations of discussions on this issue, we thought we would mainly like to study the mitigation of lipofuscin accumulation by SH-treated. Therefore, we conclude that the accumulation of lipofuscin in senescent nematodes of the same age is slowed down by SH treatment (highlighted in line 205-207). Again, we apologize for having caused you distress by not elaborating on this content.

Question 9: Line 227: It should be ‘heat shock’.

Respond: Thank you for these corrections, we have corrected it (highlighted in line 220), and the revised manuscript has been normalized to ‘heat shock’.

Question 10: Fig. 5. Figure legend is incomplete. What ‘a’, ‘b’, ‘c’ etc. means on top of the bar graphs?

Respond: Thank you for the careful reminder, we have added figure legend (highlighted in line 276-278) of Figure 5 in our revised manuscript.

Finally, thank you once again for your professional guidance and kindness!

Round 2

Reviewer 1 Report

The manuscript is sufficiently improved to be published in Foods.

Author Response

Thank you very much for your support and kindness. We will definitely keep on going!